# Characterisation and potential for reducing optical resonances in FTIR spectrometers of the Network for the Detection of Atmospheric Composition Change (NDACC)

Thomas Blumenstock[1], Frank Hase[1], Axel Keens[2], Denis Czurlok[2], Orfeo Colebatch[3], Omaira Garcia[4], David W. T. Griffith[5], Michel Grutter[6], James W. Hannigan[7], Pauli Heikkinen[8], Pascal Jeseck[9], Nicholas Jones[5], Rigel Kivi[8], Erik Lutsch[3], Maria Makarova[10], Hamud Kh. Imhasin[10], Johan Mellqvist[11], Isamu Morino[12], Tomoo Nagahama[13], Justus Notholt[14], Ivan Ortega[7], Mathias Palm[14], Uwe Raffalski[15], Markus Rettinger[16], John Robinson[17], Matthias Schneider[1], Christian Servais[18], Dan Smale[17], Wolfgang Stremme[6], Kimberly Strong[3], Ralf Sussmann[16], Yao Té[9], Voltaire A. Velazco[5]

[1]Karlsruhe Institute of Technology (KIT), Institute of Meteorology and Climate Research (IMK-ASF), Karlsruhe, Germany
[2]Bruker Optics GmbH, Ettlingen, Germany
[3]Department of Physics, University of Toronto, Toronto, Canada
[4]Izaña Atmospheric Research Centre (IARC), Meteorological State Agency of Spain (AEMET), Tenerife, Spain
[5]Centre for Atmospheric Chemistry, University of Wollongong, Wollongong, Australia
[6]Centro de Ciencias de la Atmósfera, Universidad Nacional Autónoma de México (UNAM), Mexico City, México
[7]National Center for Atmospheric Research (NCAR), Boulder, CO, USA
[8]Finnish Meteorological Institute (FMI), Sodankylä, Finland
[9]Laboratoire d'Etudes du Rayonnement et de la Matière en Astrophysique et Atmosphères (LERMA-IPSL), Sorbonne Université, CNRS, Observatoire de Paris, PSL Université, 75005 Paris, France
[10]Saint Petersburg State University, Atmospheric Physics Department, St. Petersburg, Russia
[11]Department of Earth and Space Science, Chalmers University of Technology, Göteborg, Sweden
[12]National Institute for Environmental Studies (NIES), Tsukuba, Ibaraki 305-8506, Japan
[13]Institute for Space-Earth Environmental Research (ISEE), Nagoya University, Nagoya, Japan
[14]Institute of Environmental Physics, University of Bremen, Bremen, Germany
[15]Swedish Institute of Space Physics (IRF), Kiruna, Sweden
[16]Karlsruhe Institute of Technology, IMK-IFU, Garmisch-Partenkirchen, Germany
[17]National Institute of Water and Atmospheric Research Ltd (NIWA), Lauder, New Zealand
[18]Institut d'Astrophysique et de Géophysique, Université de Liège, Liège, Belgium

*Correspondence to*: Thomas Blumenstock (thomas.blumenstock@kit.edu)

**Abstract.** Although optical components in Fourier transform infrared (FTIR) spectrometers are preferably wedged, in practice, infrared spectra typically suffer from the effects of optical resonances ("channeling") affecting the retrieval of weakly absorbing gases. This study investigates the level of channeling of each FTIR spectrometer within the Network for the Detection of Atmospheric Composition Change (NDACC). Dedicated spectra were recorded by more than twenty NDACC FTIR spectrometers using a laboratory mid-infrared source and two detectors. In the InSb detector domain (1900 – 5000 cm$^{-1}$), we find that the amplitude of the most pronounced channeling frequency amounts to 0.1 to 2.0 ‰ of the spectral background level, with a mean of (0.68 ± 0.48) ‰ and a median of 0.60 ‰. In the HgCdTe detector domain (700 – 1300 cm$^{-1}$), we find

even stronger effects, with the largest amplitude ranging from 0.3 to 21 ‰ with a mean of (2.45 ± 4.50) ‰ and a median of 1.2 ‰. For both detectors, the leading channeling frequencies are 0.9 and 0.11 or 0.23 cm$^{-1}$ in most spectrometers. The observed spectral frequencies of 0.11 and 0.23 cm$^{-1}$ correspond to the optical thickness of the beam splitter substrate. The 0.9 cm$^{-1}$ channeling is caused by the air gap in between the beam splitter and compensator plate.. Since the air gap is a significant source of channeling and the corresponding amplitude differs strongly between spectrometers, we propose new beam splitters with the wedge of the air gap increased to at least 0.8°. We tested the insertion of spacers in a beam splitter's air gap to demonstrate that increasing the wedge of the air gap decreases the 0.9 cm$^{-1}$ channeling amplitude significantly. A wedge of the the air gap of 0.8° reduces the channeling amplitude by about 50% while a wedge of about 2° removes the 0.9 cm$^{-1}$ channeling completely. This study shows the potential for reducing channeling in the FTIR spectrometers operated by the NDACC, thereby increasing the quality of recorded spectra across the network.

## 1 Introduction

Ground-based FTIR (Fourier transform infrared) spectroscopy is a widely used technique for measuring total and partial column abundances of a variety of trace gases in the atmosphere. Within the Network for the Detection of Atmospheric Composition Change (NDACC), this technique is used at about twenty sites covering a wide range of geographical latitudes. The NDACC data are used to study short and long-term variability of the atmosphere as well as for satellite data validation (De Mazière et al., 2018). For both applications, high data quality and station-to-station consistency are of utmost importance. Ground-based FTIR spectroscopy provides data of high quality (e.g. Schneider and Hase, 2008). However, several key instrumental characteristics need to be addressed. These parameters such as detector non-linearity (Abrams et al., 1994), instrumental line shape (ILS; Hase et al., 1999), intensity fluctuations (Keppel-Aleks et al., 2007), precise solar tracking (Gisi et al., 2011), and sampling error (Messerschmidt et al., 2010; Dohe et al., 2013) have been studied in some detail and need to be taken into account.

In this paper, channeling – the presence of instrument-induced periodic oscillations of spectral transmission resulting from internal optical resonances – will be investigated and discussed. In the past, each site or each new spectrometer was tested for channeling individually. This paper describes a network-wide exercise for characterizing channeling in FTIR spectrometers. Channeling is caused by interference of reflections of the incoming light at parallel transmitting surfaces of optical elements. In practice, the resulting channeling amplitudes are less than 10 ‰ in signal. Thus, the retrieved data for species with strong absorption signatures, as for example ozone and many others, are less critically affected. However, the retrieved trace gas amounts of weak absorbers can be substantially disturbed. In such cases, channeling becomes an important component of the total error budget.

Recently, time series of column abundances of formaldehyde (HCHO) were retrieved from NDACC FTIR sites (Vigouroux et al., 2018, 2020). The studies of Vigouroux also includes an error characterisation of the HCHO product. Within the network, two retrieval codes are in use: SFIT4 and PROFFIT. While the retrieval codes were inter-compared and show consistent results

(Hase et al., 2004), the assumed error budgets differ slightly. The stations using PROFFIT include an error contribution due to channeling while the stations using SFIT4 do not. The result is a larger total error for HCHO data retrieved with PROFFIT as compared to SFIT4 (Vigouroux et al., 2018). In the PROFFIT error calculation, a set of typical channeling frequencies and amplitudes is taken into account. More specifically, channeling amplitudes of 0.5 ‰ for four frequencies are assumed: 0.005,

0.2, 1.0, and 3.0 cm$^{-1}$. The resulting error contribution doubles the total error of HCHO columns amounts.

In order to make this assumption more robust and to quantify more carefully the differences from spectrometer to spectrometer, an exercise was performed to measure channeling frequencies and amplitudes of NDACC FTIR spectrometers. Since atmospheric spectra are densely populated with absorption signatures interfering with the signal generated by channeling; the test was designed using spectra collected in a laboratory setting.  Section 2 briefly describes the origin of channeling, Sect. 3

the setup of this exercise, and Sect. 4 shows the results followed by a discussion. Finally, to reduce the channeling amplitude, the investigation of a modified beam splitter design is presented in Sect. 5, and lastly, Sect. 6 gives the conclusions.

## 2 Spectral transmission of a Fabry-Perot cavity

In an FTIR spectrometer, the transmitted light passes through several optical components such as optical windows, optical

filters and a beam splitter, typically comprised of a beam-splitting layer system deposited on a transparent substrate and a compensator. At the transmitting surfaces of these components, the optical beam is partially reflected. In the case of parallel surfaces, each pair of surfaces defines a cavity (Fig. 1a) in which multiple reflections occur. Due to interference of the reflected light, a standing wave is created (Fig. 1b). This effect is called the Fabry-Perot or etalon effect or channeling. The optical length of the cavity defines the free spectral range $\nu_{(FSR)}$ as

$$\nu(FSR) = 1/(2nd\cos\theta) \tag{1}$$

with $n$ refractive index and $d$ thickness of the optical component (Hecht, 2017). $\theta$ is the angle between incoming light beam and the normal of the optical surface (Fig. 1a). Equation (1) is used to identify the optical element responsible for a certain channeling frequency. Table 1 gives a few examples of $\nu_{(FSR)}$ for optical materials commonly used in FTIR spectrometers.


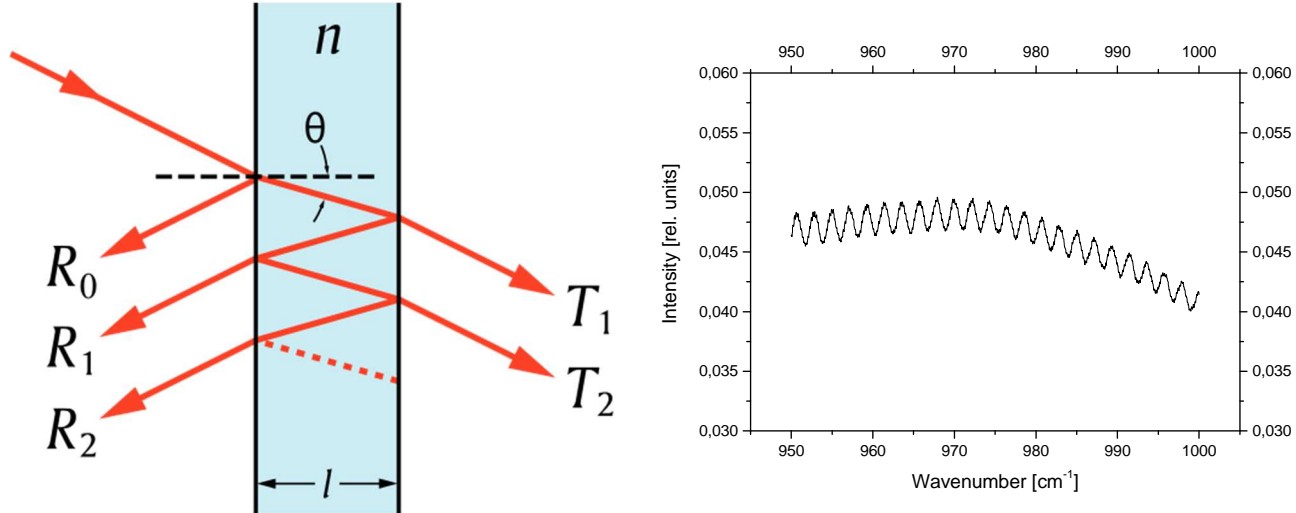

**Figure 1:** (a) Multiple reflections at parallel surfaces in an optical component where 'l' is denoted 'd' in equation (1) (taken from Wikimedia Commons: https://commons.wikimedia.org/), (b) Channeling in an IR spectrum.

**Table 1:** Free spectral range $\nu_{(FSR)}$ of some components typically used in NDACC FTIR spectrometers with $\cos \theta = 1$.

| Material | used as | $n$ | $d$ [mm] | $\nu_{(FSR)}$ [cm$^{-1}$] |
|---|---|---|---|---|
| Air | Gap in between beam splitter and compensator plate | 1 | 5.5 | 0.91 |
| KBr | Beam splitter substrate | 1.5 | 15 | 0.22 |
| CaF$_2$ | Beam splitter substrate | 1.4 | 15 | 0.24 |
| CaF$_2$ | Detector window | 1.4 | 1.0 | 3.57 |
| Ge | Detector window | 4.4 | 1.0 | 1.14 |
| KRS-5 (TlBr-TlI) | Detector window | 2.37 | 1.0 | 2.11 |
| Sapphire | Detector window | 1.65 | 1.0 | 3.0 |
| ZnSe | Detector window | 2.2 | 1.0 | 2.27 |

The Fabry-Pérot etalons generated by these optical components have rather low reflectivity and therefore the undesired parasitic effects caused in their spectral transmission is well described as a harmonic oscillation. For demonstrating the plausibility of our empirical experimental results, we here provide some basic considerations concerning the channeling effects created by a Fabry-Perot etalon of low finesse. Further background information can be found in Ismail et al., (2016) and references herein.


### 2.1 Fabry-Perot effect in a plane-parallel window at normal incidence

Assume a plane-parallel KBr window of thickness $d$ at normal incidence. The refractive index of KBr is 1.5346 at 5 μm and 1.5265 at 10 μm (see https://refractiveindex.info/?shelf=main&book=KBr&page=Li and references therein). We here assume a low finesse, so higher order contributions to the modulated transmission can be neglected. The channeling results from the

superposition of the primary transmitted beam with a parasitic beam which is generated by reflection at the exit surface (as result, travelling in the opposite direction as the primary beam) and afterwards at the entrance surface (as result, being redirected again, travelling again parallel to the primary beam). The ratio of intensities between the parasitic and primary beams is given by the Fresnel relation for normal rays:

$$R = \left|\frac{m-1}{m+1}\right|^2 \tag{2}$$

Here, $m$ is the ratio of the refractive indices involved (here, those of KBr and vacuum or air $n\_air = 1.00027 \approx 1$). Because the parasitic ray undergoes two reflections, the intensity ratio is 1.979 ‰ at 5 μm and 1.886 ‰ at 10 μm. This requires that the ratio of the electric amplitudes of the monochromatic electromagnetic waves represented by the two beams is the square root of these values, so 0.0445 at 5 μm and 0.0434 at 10 μm. From a vector addition of the electric amplitudes of the primary and the parasitic ray the peak-to-peak amplitude of the channeling follows: it amounts to a peak-to-peak variation in the intensity

of 178 ‰ at 5 μm and 174 ‰ at 10 μm (note that the channeling signal is detected by measuring variable intensities, not wave amplitudes).

The periodicity of the channeling is determined by the requirement that for constructive interference, the path difference between the primary and the parasitic ray needs to equal the extra optical path length travelled by the parasitic ray:

$$2nd = N\lambda \tag{3}$$

Here, $n$ is the refractive index of the plate, $\lambda$ is the vacuum wavelength, and $N$ is a positive integer number. By rearranging the equation for representation as a function of wavenumbers we find that the fringe period $\Delta v$ becomes equidistant as function of wavenumber if the refractive index is constant. If we allow for dispersion $n = n(v)$, the channeling period of eq. 1 becomes slightly wavenumber dependent.

$$\Delta v = \frac{1}{2n(v)d} \tag{4}$$

Note that a resonator formed by a gap instead of KBr will show no (in vacuum) or much less (in laboratory air) variability of the fringe period.

## 2.2 Fabry-Perot effect in a plane-parallel KBr plate at 30° angle of incidence

Now we investigate a plane-parallel KBr plate of thickness $d$ at 30° angle of incidence, the typical angle in the Bruker FTIR
systems. The intensities of the primary and parasitic beams now depend on the state of polarisation. The Fresnel relations for oblique rays provide the reflectivities for linearly polarized waves with the $E$ vector oscillating in the plane of incidence ($R_p$) or perpendicular to it ($R_S$):

$$R_P = \left| \frac{\cos\beta - m\cos\alpha}{\cos\beta + m\cos\alpha} \right|^2 \qquad \text{and} \qquad R_S = \left| \frac{\cos\alpha - m\cos\beta}{\cos\alpha + m\cos\beta} \right|^2 \tag{5}$$

Here, $\alpha$ is the incidence angle, while $\beta$ is the angle with respect to the normal inside the plate. For 30° incidence angle (so
$\beta = 19.02°$ at 5 µm and $\beta = 19.12°$ at 10 µm), we calculate the reflectivities as provided in Table 2.

**Table 2:** Reflectivities calculated from the Fresnel relations

| Wavelength [µm] | $R_P$ | $R_S$ |
|---|---|---|
| 5 | 0.02845 | 0.06371 |
| 10 | 0.02768 | 0.06232 |

While $R_p$ decreased in comparison to the reflectivity for normal incidence ($\approx 0.04$), the value of $R_S$ increased. Note that under
the Brewster angle, $R_p$ would vanish and channeling caused by the beam splitter (BS) could be removed completely. Operation of a BS near the Brewster angle (here $\approx 57°$) and introduction of a polarizing unit selecting only the perpendicular component for detection would in principle be an alternative approach for removing channeling generated by the BS. However, this would require a complete re-design of the spectrometer setup (using the BS at a rather inconvenient angle of incidence of nearly 60°) and it would reduce the amount of signal if the source provides unpolarised radiation. (However, the significant polarisation-
dependency of the channeling following from the Fresnel equations could be used to prove whether a channeling fringe is created by the BS by using a polarisation filter in front of the detector). Here, if we work with an unpolarised source, we can assume that the channeling amplitude will not be very different from the amplitude estimated for normal incidence.

The period of the channeling fringe as function of wavenumber becomes shorter for geometric reasons when the plate orientation is tilted away from normal incidence: the effective thickness of the BS increases. Note that the change of the
channeling period in the presence of dispersion now is created by two mechanisms: the changing relation between optical and geometric path length and the changing angle of transmission:

$$\Delta\nu = \frac{\cos\beta}{2n(\nu)d} \tag{6}$$

**2.3 Fabry-Perot effect in a wedged plate**

We have seen that there is no significant impact of wavenumber on the channeling amplitude for a plane parallel plate. We will, however, show that a wedge of certain amount is significantly more effective in suppressing channeling at shorter wavelengths.

For our investigation, we assume that the source is incoherent. Therefore, the primary beam can only interfere with the parasitic beam deviated by the wedge (not with a parasitic beam emerging from a different position in the source and exiting the BS

under the same angle as the primary beam). As result of the wedge, the wave front of the parasitic beam is now tilted with respect to the primary beam. We analyse the resulting effect on the circular aperture of the collimator focusing the radiation emerging from the interferometer on the exit aperture. The tilt between the outgoing wave fronts of the primary and parasitic plane waves generates equidistant straight stripes of constant phase shift in that plane (stripe orientation perpendicular to wedge). What has been a uniform variation of brightness across the collimator aperture (when either tuning wavelength or

plate thickness) now becomes a shift of the stripe pattern perpendicular to the orientation of the stripes. We can estimate the damping effect introduced by the wedge by determining the residual brightness fluctuations emerging from the shifting stripe pattern (technically by integration over the aperture). Obviously, if the stripe pattern becomes denser (larger wedge or shorter wavelength), the brightness fluctuations are further and further reduced. Figure 2 shows the amplitude of the integrated brightness fluctuation as function of cycles across the aperture of the collimator (each cycle is equivalent to adding a detuning

of one wavelength across the aperture of the collimator), given by

$$ncycles = \nu D \sin(2\omega) \tag{7}$$

Here, $\nu$ is the wavenumber, D the beam diameter, and $\omega$ the wedge angle.

Note that our consideration shows that the channeling amplitude is reduced when (1) the aperture of the collimator (or, equivalently, the beam diameter supported by the interferometer) is increased (2) the wavelength is reduced, or (3) the wedge

angle is increased.

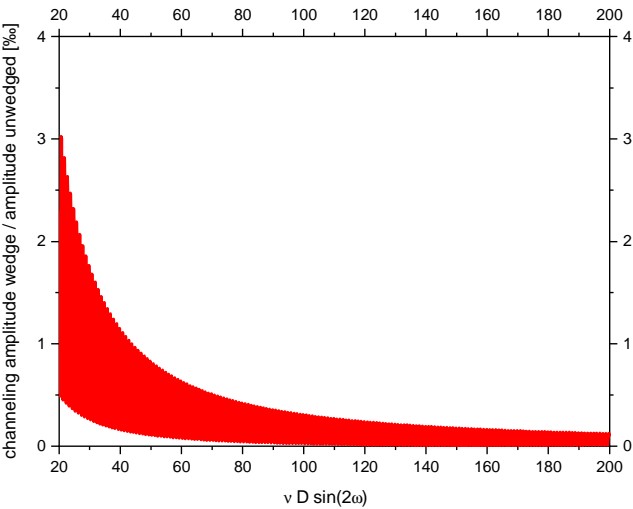

Figure 2: Channeling amplitude as function of wedge angle

While a Fabry-Pérot spectrometer is designed and aligned such that the surfaces are parallel to build a cavity, an FTIR
      spectrometer is designed differently: In order to reduce or avoid channeling, optical components need to be wedged or installed
      with a large tilt. A large tilt is not feasible in many cases. Thus, optical components are normally wedged. As shown in this
      section wedged optical components reduce channeling because the reflected beams do not superimpose and thus, do not
      interfere with each other. These wedged components require a special design and limits compatibility with non-wedged
devices. Furthermore, some components such as detector elements are not available as wedged versions (the partially
      transparent detector element can also act as optical cavity). Therefore, in practice it is challenging to build an FTIR
      spectrometer that is completely free of channeling.

### 3 Channeling test exercise

### 3.1 Experimental setup

In atmospheric spectra, channeling can be difficult to see due to the presence of complex atmospheric signatures. Therefore,
      laboratory spectra are used for this exercise, recorded either with a mid-infrared globar or with a black body of at least 1000 °C
      temperature. Since these types of sources do not include a window, no additional channeling is added to the spectra. A
      temperature of 1000 °C is required to record spectra with a sufficient signal-to-noise ratio in a reasonable amount of time.
      Within NDACC, two detectors and the NDACC filter set are used (Table A1). The optical filters are used to increase the signal
to noise ratio of the spectra. The NDACC filters have a wedge of 0.17° and therefore, if properly oriented, do not cause

channeling. Therefore, not all filters but both detectors were included in this exercise. More specifically, NDACC filter #3 (2400 to 3000 cm$^{-1}$ spectral range) for the InSb detector and NDACC filter #6 (700 to 1300 cm$^{-1}$ spectral range) for the HgCdTe detector were used. Some sites (Harestua, Paris, Wollongong, and Lauder_120HR) use filter #7 (700 to 1000 cm$^{-1}$ spectral range) and #8 (1000 to 1400 cm$^{-1}$ spectral range) instead of filter #6 (Table 3). In this case, filter #7 was used for this exercise. Filter #3 was selected since this filter range is used for the retrieval of HCHO column abundances.

Multiple reflections within optical components such as optical windows or beam splitters typically show channeling frequencies of a few tenths of a wavenumber up to a few wavenumbers. In general, higher frequency channeling with wavenumbers below 0.1 cm$^{-1}$ might occur when different optical components form the surfaces of the resulting cavity, e.g. in the Bruker 120HR spectrometer the rim of the entrance field stop is part of a resonator of about 1 m length. However, this is seldom the case in an FTIR spectrometer and secondly, due to the high frequency, easily detectable even in atmospheric spectra.

In order to focus on channeling due to multiple reflections inside optical components and to achieve a very good signal-to-noise ratio, a spectral resolution of 0.05 cm$^{-1}$ (OPD = 180 cm) was chosen. This resolution allowed us to add thousand interferograms within a few hours, thereby achieving signal-to-noise ratio that allowed channeling amplitudes to be detected and quantified on a per mille scale.

### 3.2 Analysis of channeling test spectra

To quantify channeling frequencies and their amplitudes, an FFT (Fast Fourier Transform) analysis of the spectra was conducted. First of all, a spectral interval was chosen with a nearly constant intensity: 950 to 1000 cm$^{-1}$ for HgCdTe and 2550 to 2600 cm$^{-1}$ for InSb spectra. This step was carried out using OPUS™, a software package from Bruker Optics to control FTIR spectrometers (Fig. 3a). Then, the background was normalized by dividing a straight line that connects the ends of the spectrum using ORIGIN™ software (red line in Fig.3a). The quotient minus 1 is the basis for the FFT analysis (Fig. 3b). Finally, an inverse FFT was conducted also with Origin™ software (Fig. 3c).

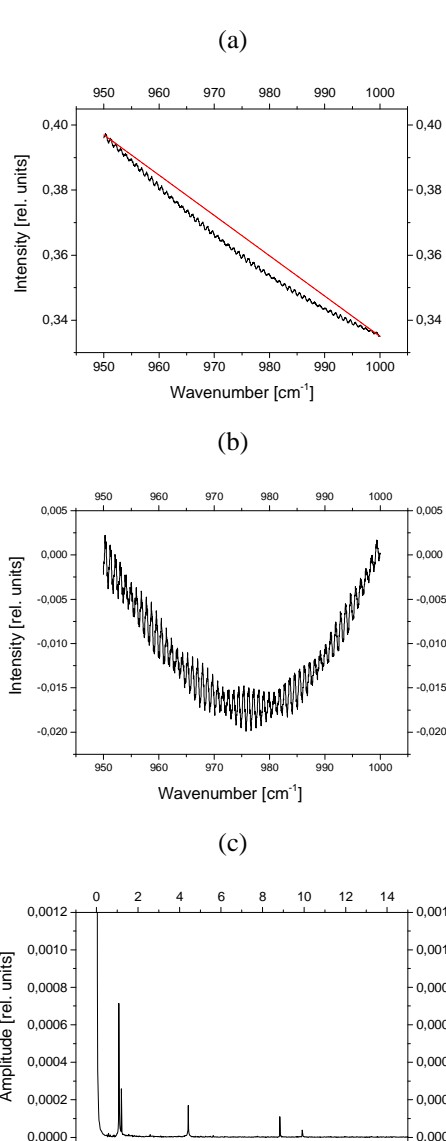

(a)

(b)

235

(c)

**Figure 3:** Analysis of a channeling test spectrum: (a) Cut off a window of 50 cm$^{-1}$; a straight line is calculated that connects the ends of the spectrum (red line); (b) Normalize background by dividing this straight line and subtract a constant of 1; (c) Result of FFT analysis

240 **4 Results and Discussion**

In this section, the results are presented for more than twenty spectrometers. Table 3 provides the list of spectrometers included in this study. Please note that a few spectrometers do not include an HgCdTe detector: Garmisch, Karlsruhe, and Sodankylä.

**Table 3**: List of spectrometers contributing to the channeling test exercise, sorted by latitude of the site, from north (Eureka) to south (Arrival Heights).

| Site | Acronym | Type | Beam splitter setup | Optical filter | Team |
|------|---------|------|---------------------|----------------|------|
| Eureka | EUR | Bruker 125 HR | KBr | #3 & #6 | U Toronto |
| Ny-Ålesund | NY | Bruker 120/5 HR | KBr for HgCdTe, CaF$_2$ for InSb det. | #3 & #6 | U Bremen |
| Thule | THU | Bruker 125 HR | KBr | #3 | NCAR |
| Kiruna | KIR | Bruker 120/5 HR | KBr | #3 & #6 | KIT-ASF, IRF |
| Sodankylä | SOD | Bruker 125 HR | CaF$_2$, no HgCdTe det. | #3 | FMI |
| Harestua | HAR | Bruker 120 M | KBr | #3 & #8 | U Gothenborg |
| St. Petersburg | STP | Bruker 120 HR | KBr | Ind. #3 & #6 | SPbU |
| Bremen | BRE | Bruker 125 HR | KBr | #3 & #6 | U Bremen |
| Karlsruhe | KAR | Bruker 125 HR | CaF$_2$, no HgCdTe det. | #3 | KIT-ASF |
| Paris | PAR | Bruker 125 HR | KBr for HgCdTe, CaF$_2$ for InSb det. | Ind. #3 & #7 | Sorbonne U |
| Garmisch | GAR | Bruker 125 HR | CaF$_2$, no HgCdTe det. | #3 | KIT-IFU |
| Zugspitze | ZUG | Bruker 120/5 HR | KBr | #3 & #6 | KIT-IFU |
| Jungfraujoch | JJO | Bruker 120 HR | KBr | #3 & #6 | U Liège |
| Toronto | TOR | BOMEM DA8 | KBr | #3 & #6 | U Toronto |
| Rikubetsu | RIK | Bruker 120/5 HR | KBr for HgCdTe, CaF$_2$ for InSb det. | #3 & #6 | U Nagoya, NIES |
| Boulder | BOU | Bruker 120/5 HR | KBr | #3 | NCAR |
| Tsukuba | TSU | Bruker 125 HR | KBr for HgCdTe, CaF$_2$ for InSb det. | #3 & #6 | NIES |
| Izaña | IZ | Bruker 120/5 HR | KBr | #3 & #6 | AEMet, KIT-ASF |
| Mauna Loa | MLO | Bruker 120/5 HR | KBr | #3 & #6 | NCAR |
| Altzomoni | ALT | Bruker 120/5 HR | KBr | #3 & #6 | UNAM |
| Wollongong | WOL | Bruker 125 HR | KBr | #3 & #7 | U Wollongong |
| Lauder | LAU | Bruker 120 HR / Bruker 125 HR | KBr / KBr | #3 & #7 / #3 & #6 | NIWA |
| Arrival Heights | AH | Bruker 125 HR | KBr | #3 & #6 | NIWA |

These sites primarily serve the TCCON (Total Carbon Column Observing Network; Wunch et al., 2010) and just contribute with InSb spectra to NDACC and to this exercise. These spectrometers use a CaF$_2$ beam splitter instead of KBr; the latter is normally used in NDACC for enabling measurements in the HgCdTe spectral range. Ny-Ålesund, Paris, Rikubetsu and Tsukuba sites use a CaF$_2$ beam splitter for InSb and a KBr beam splitter for HgCdTe measurements. Tables 4 and 5 list the detected channeling frequencies and their amplitudes in spectra recorded with InSb and HgCdTe detectors, respectively.

### 4.1 InSb detector domain

Figure 4 shows the detected channeling frequencies and their amplitudes in InSb spectra analysed at about 2600 cm$^{-1}$. Most spectrometers show the expected channeling frequencies: about 0.9 cm$^{-1}$ and 0.11 or 0.23 cm$^{-1}$. These frequencies are consistent with (i) the gap between beam splitter and compensator plate (0.9 cm$^{-1}$), and (ii) the beam splitter substrate (0.23 cm$^{-1}$; Table 1). A frequency of 0.11 cm$^{-1}$ corresponds to a resonator due to both substrates, the beam splitter and the compensator plate. A few spectrometers (Harestua, Garmisch, Toronto, Boulder and Izaña-2018) show an additional channeling fringe with a frequency of about 3 cm$^{-1}$. This is due to the detector window that is often made of sapphire or calcium fluoride (CaF$_2$). Also in Izaña, this channeling frequency was detected in 2018. In December 2018, the detector was exchanged because of decreasing sensitivity. The new detector (Izaña-2019) shows much less channeling. Detectors purchased in the 1990s sometimes had a detector window with insufficient wedge.

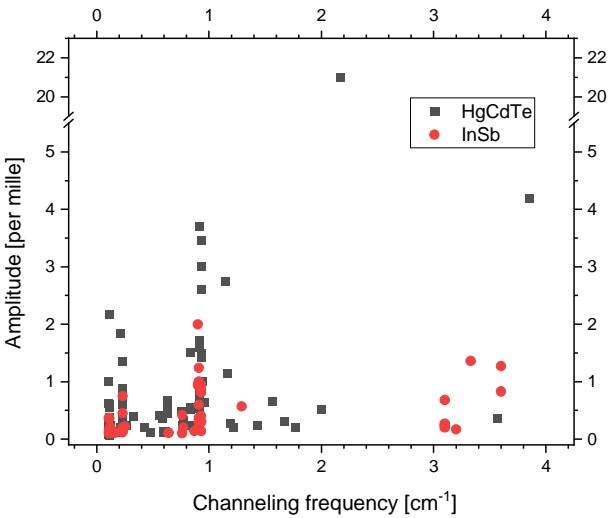

**Figure 4:** Amplitude of channeling frequencies as observed in InSb and HgCdTe test spectra

**Table 4:** Leading channeling frequencies F and their amplitudes A in the InSb detector regime. Channeling amplitudes larger than 0.6 ‰ are highlighted in bold.

| FTIR site | F 1 [cm⁻¹] | A 1 [‰] | F 2 [cm⁻¹] | A 2 [‰] | F 3 [cm⁻¹] | A 3 [‰] | F 4 [cm⁻¹] | A 4 [‰] |
|---|---|---|---|---|---|---|---|---|
| Eureka | 0.93 | 0.14 | 0.23 | 0.05 | 0.11 | 0.004 | | |
| Ny-Ålesund | **0.90** | **2.0** | 0.11 | 0.08 | | | | |
| Thule | **0.91** | **1.0** | 0.23 | 0.18 | 0.11 | 0.15 | 3.1 | 0.27 |
| Kiruna | 0.85 | 0.05 | 0.11 | 0.003 | 0.76 | 0.1 | | |
| Sodankylä | 0.93 | 0.3 | 0.12 | 0.03 | 0.11 | 0.024 | 0.25 | 0.01 |
| Harestua | 0.91 | 0.37 | 0.10 | 0.02 | **3.33** | **1.36** | | |
| St. Petersburg | 0.93 | 0.3 | 0.23 | 0.12 | 0.16 | 0.11 | 0.77 | 0.20 |
| Bremen | 0.93 | 0.3 | 0.23 | 0.16 | 0.11 | 0.05 | | |
| Karlsruhe | 0.87 | 0.14 | | | 1.29 | 0.57 | | |
| Paris | 0.91 | 0.2 | 0.25 | 0.05 | | | | |
| Garmisch | 0.91 | 0.6 | 0.10 | <0.1 | 3.1 | 0.24 | | |
| Zugspitze | 0.91 | 0.26 | 0.11 | 0.025 | 0.10 | 0.035 | | |
| Jungfraujoch | **0.91** | **1.24** | 0.23 | 0.08 | 0.12 | 0.02 | | |
| Toronto | **3.10** | **0.68** | 0.21 | 0.05 | 0.11 | 0.02 | | |
| Rikubetsu | **0.90** | **0.94** | 0.25 | 0.22 | 0.11 | 0.11 | 3.2 | 0.17 |
| Boulder | **0.93** | **0.81** | **0.23** | **0.75** | 0.11 | 0.11 | 3.6 | 0.83 |
| Tsukuba | **0.93** | **0.94** | 0.12 | 0.21 | 0.11 | 0.10 | | |
| Izaña – 2018 | 0.76 | 0.42 | 0.10 | 0.09 | 0.11 | 0.06 | **3.6** | **1.27** |
| Izaña – 2019 | 0.83 | 0.07 | 0.10 | 0.02 | 0.11 | 0.03 | 3.1 | 0.20 |
| Mauna Loa | **0.93** | **0.85** | 0.23 | 0.45 | 0.11 | 0.36 | | |
| Altzomoni | 0.64 | 0.11 | 1.82 | 0.04 | 0.74 | 0.03 | | |
| Wollongong | 0.93 | 0.40 | 0.23 | 0.20 | 0.11 | 0.03 | | |
| Lauder 120HR | 0.91 | 0.32 | 0.23 | 0.08 | 0.11 | 0.02 | | |
| Lauder125HR | **0.91** | **1.0** | 0.23 | 0.14 | 0.11 | 0.37 | 0.10 | 0.06 |
| Arrival Heights | **0.91** | **0.94** | 0.23 | 0.03 | 0.12 | 0.11 | 0.10 | 0.09 |

270

**Table 5:** Leading channeling frequencies F and their amplitudes A in the HgCdTe detector regime. Channeling amplitudes larger than 1.2 ‰ are printed in bold.

| FTIR site | F 1 [cm⁻¹] | A 1 [‰] | F 2 [cm⁻¹] | A 2 [‰] | F 3 [cm⁻¹] | A 3 [‰] | F 4 [cm⁻¹] | A 4 [‰] |
|---|---|---|---|---|---|---|---|---|
| Eureka | **0.93** | **1.5** | 0.23 | 0.2 | 0.11 0.10 | 0.14 0.05 | | |
| Ny-Ålesund | **0.91** | **1.6** | 0.23 **0.21** | 0.89 **1.85** | 0.11 0.10 | 0.60 0.62 | **2.17** | **21** |
| Kiruna | 0.77 | 0.32 | 0.59 | 0.12 | 0.11 | 0.07 | | |
| Harestua | **0.91** | **3.7** | 0.23 0.11 | 0.73 0.16 | 1.56 0.58 | 0.66 0.36 | **3.85** | **4.2** |
| St. Petersburg | 0.94 | 1.0 | 0.23 0.33 | 0.30 0.40 | 2.0 1.77 | 0.52 0.20 | | |
| Bremen | **0.93** 0.83 | **1.43** 0.52 | 0.23 | 0.34 | 0.11 0.10 | 0.22 0.08 | | |
| Paris | 0.83 | 0.56 | 0.26 0.23 | 0.23 0.37 | 0.21 0.12 | 0.13 0.23 | | |
| Zugspitze | 0.91 | 0.79 | 0.23 | 0.25 | 0.11 0.10 | 0.18 0.19 | 3.57 | 0.36 |
| Jungfraujoch | 0.91 | 0.53 | 0.23 0.21 | 0.60 0.12 | 0.11 0.10 | 0.17 0.06 | | |
| Toronto | 0.96 0.48 | 0.64 0.12 | 0.21 | 0.20 | 0.10 | 0.10 | | |
| Rikubetsu | **0.93** **0.83** | **1.44** **1.51** | 0.23 0.18 | 0.62 0.14 | **0.11** 0.10 | **2.18** 1.01 | 0.42 | 0.21 |
| Tsukuba | **0.93** | **3.46** | 0.23 | 0.67 | 0.11 0.10 | 0.38 0.33 | 1.19 | 0.27 |
| Izaña – 2018 | 0.76 | 0.23 | 0.63 0.56 | 0.45 0.41 | 0.11 0.10 | 0.13 0.13 | | |
| Izaña – 2019 | 0.75 | 0.48 | 0.63 | 0.54 | 0.11 | 0.17 | | |
| Mauna Loa | **0.93** | **2.60** | **0.23** | **1.35** | 0.11 0.10 | 0.56 0.10 | 0.61 | 0.14 |
| Altzomoni | 0.88 0.63 | 0.25 0.68 | 1.67 1.43 | 0.31 0.23 | 0.11 | 0.08 | 1.22 | 0.21 |
| Wollongong | **0.93** 0.82 | **3.00** 0.23 | 0.23 0.59 | 0.25 0.13 | 0.11 | 0.16 | | |
| Lauder 120HR | 0.91 | 0.72 | 0.23 | 0.06 | 0.11 0.10 | 0.12 0.07 | 1.51 | 0.08 |
| Lauder 125HR | **0.91** | **1.69** | 0.23 | 0.41 | 0.11 0.10 | 0.23 0.11 | **1.14** | **2.74** |
| Arrival Heights | **0.91** | **1.72** | 0.23 | 0.18 | 0.11 0.10 | 0.12 0.17 | 1.16 | 1.15 |

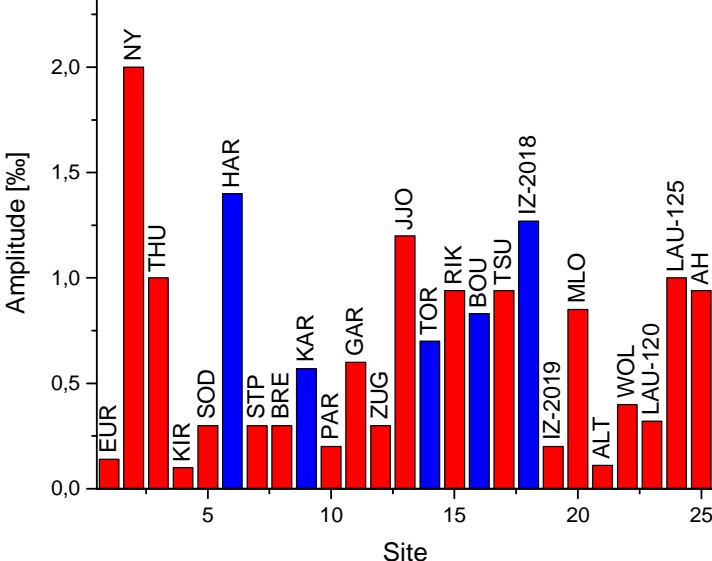

**Figure 5:** Amplitude of largest channeling fringe in test spectrum using InSb detector and NDACC filter number 3. Red bars indicate channeling due to beam splitter air gap and blue bars indicate detector window as source of channeling.

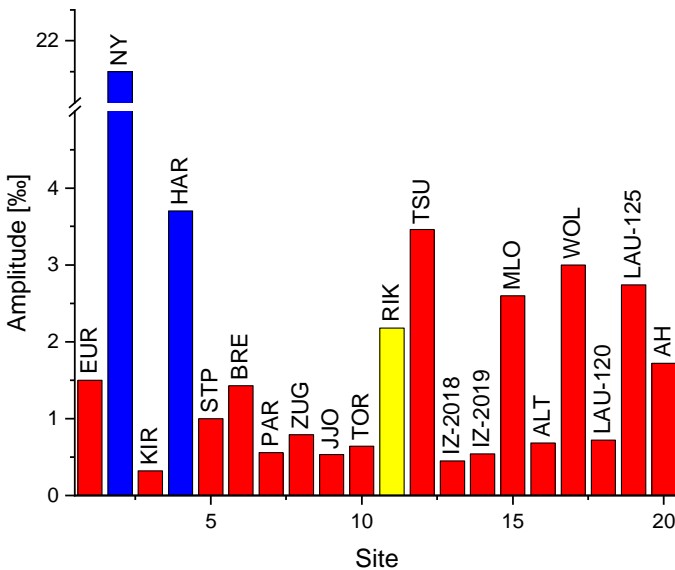

**Figure 6:** Amplitude of largest channeling fringe in HgCdTe test spectrum. Red bars indicate channeling due to beam splitter air gap, yellow bar indicates beam splitter substrate and blue bars indicate detector window as source of channeling.

Figure 5 shows the amplitude of the strongest channeling frequency of each spectrometer. The amplitudes range from 0.1 to 2.0 ‰ with a mean of (0.68 +/- 0.48) ‰ and a median of 0.60 ‰. In most cases, channeling caused by the gap of the beam splitter is the most pronounced one. These mean and median are consistent with the PROFFIT error estimate of 0.5 ‰ as used in Vigouroux et al. (2018). However, the channeling amplitude differs strongly from spectrometer to spectrometer and a few spectrometers show an amplitude of up to 21 ‰.

## 4.2 HgCdTe detector domain

Fig. 4 and Table 5 presents major channeling frequencies and their amplitudes in spectra recorded with an HgCdTe detector at about 1000 $cm^{-1}$. As for the InSb detector, most spectrometers show two dominant channeling frequencies: about 0.9 $cm^{-1}$ and 0.1 or 0.2 $cm^{-1}$ caused by the beam splitter (Table 1). Two spectrometers (Ny--Ålesund and Harestua) show an additional channeling frequency of 2.17 and 3.85 $cm^{-1}$, indicating that the wedge of the detector window is not sufficient in these cases.

Figure 6 shows the amplitude of the strongest channeling frequency of each spectrometer. The amplitudes range from 0.3 to 21 ‰ with a mean of (2.45 +/- 4.50) ‰ and a median of 1.2 ‰. In most cases, channeling caused by the gap of the beam splitter is the most pronounced one. The amplitude is even larger as compared to the InSb domain that confirms that the wedge is more efficient in reducing the channeling at shorter wavelengths as calculated in Sect. 2. At several sites, a reduction of channeling amplitudes would be desirable in order to improve trace gas retrievals of species with weak signatures, in particular from HgCdTe spectra, e.g. of $ClONO_2$, $HNO_3$ or $SF_6$.

As for the InSb domain, channeling amplitudes differ strongly from spectrometer to spectrometer. Figure 7 shows HgCdTe spectra with different levels of channeling of the same frequency (about 0.9 $cm^{-1}$) demonstrating the need of increasing the wedge of the gap and for narrowing the tolerances of wedges in the manufacturing of the beam splitters.

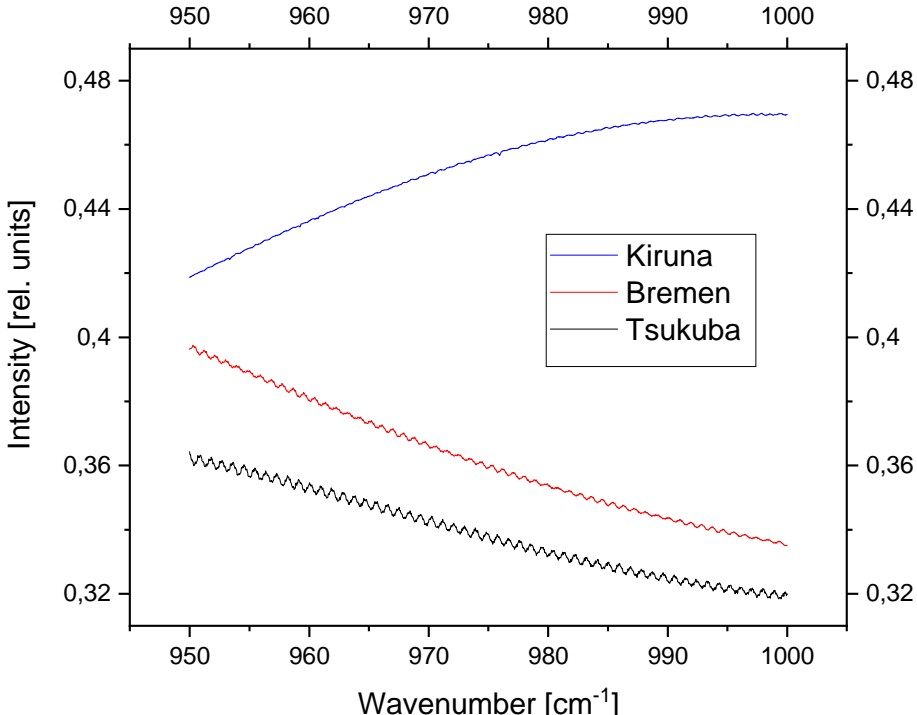

 **Figure 7:** HgCdTe spectra with low (0.32 ‰), medium (1.43 ‰) and high (3.46 ‰) channeling amplitude at 0.9 cm$^{-1}$ frequency.

## 5 Investigation of a modified beam splitter design for reducing channeling

This test exercise has found that the channeling amplitude differs strongly from spectrometer to spectrometer. A few spectrometers (at Altzomoni, Izaña, Karlsruhe and Kiruna) use customer-specific beam splitters with an increased wedge of
305    1.75° for the air gap and 0.17° for the CaF$_2$ substrate and 0.13° for the KBr substrate. Their channeling amplitudes are the lowest among all the spectrometers studied in this paper. Unfortunately, this type of beam splitter is not a standard device and is not compatible with standard beam splitters, as it requires a realignment of the interferometer. Namely due to its incompatibility with unwedged far-infrared pellicle beam splitters, the manufacturer Bruker adheres to the standard design with lower substrate wedge.

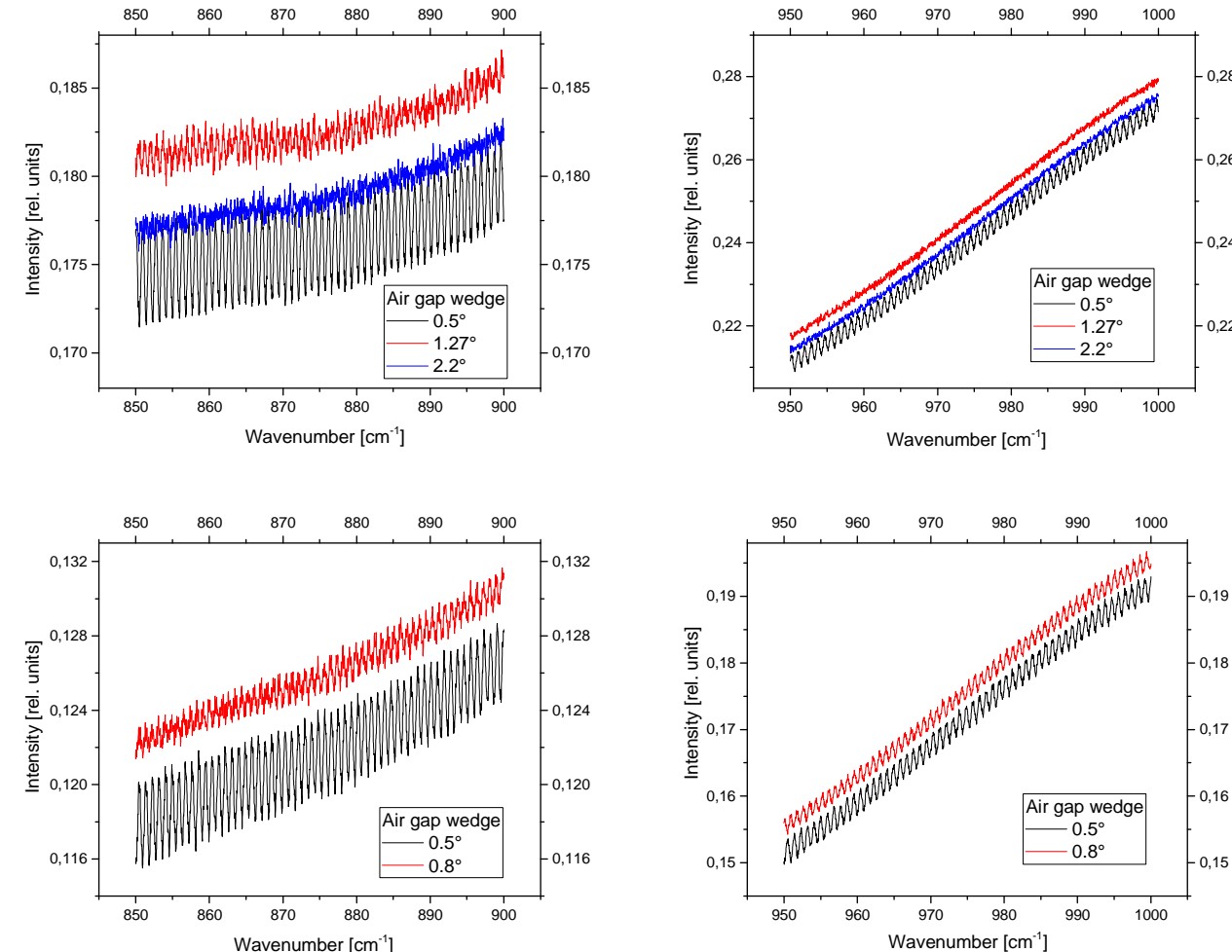

310

**Figure 8:** HgCdTe spectra recorded with different wedges of the air gap in between beam splitter and compensator plate for the 850 to 950 cm$^{-1}$ and the 950 to 1000 cm$^{-1}$ spectral ranges. These measurements were made at Bruker Company, Ettlingen,

315   using the same instrument.

To avoid the need for strongly wedged substrates, a different approach is proposed here. We focus on the wedge of the gap between the beam splitter and the compensator plate. Since the largest channeling amplitude (at 0.9 cm$^{-1}$ frequency) is caused by the air gap, an increased wedge of this gap has the potential to reduce channeling significantly. The typical air gap wedge

320   for the Bruker beam splitter is 0.5°. Different spacers with wedges of 0.5°, 1.27° and 2.2° have been manufactured by Bruker and tested. Figure 8 (upper panels) shows the resulting channeling test spectra recorded with an HgCdTe detector. Similar to most of the NDACC spectrometers, the spectrum of the 0.5° wedged beam splitter shows a pronounced channeling with an

amplitude of 5.7 ‰. In contrast, the 1.27° and 2.2° wedged beam splitters are (nearly) free of channeling with an amplitude of 0.46 and of 0.87 ‰, respectively, that is close to the noise level of these spectra . Analysed in the 850 to 900 cm$^{-1}$ spectral range, the amplitude is 8.9, 3.3 and 0.6 ‰ for a wedge of 0.5°, 1.27° and 2.2°, respectively. For InSb spectra, the 0.9 cm$^{-1}$ channeling generates amplitudes of 0.9, 0.45 and 0.19 ‰ for beam splitters with wedges of 0.5°, 1.27° and 2.2°, respectively. To ensure compatibility between different beam splitters, the wedge should be limited to 0.8°. This design will be implemented in future Bruker HR spectrometers. Figure 8 (lower panels) presents test spectra with an air gap wedge of 0.5° and 0.8°. In the 850 to 900 cm$^{-1}$ spectral range, even the slightly increased wedge reduces the channeling by nearly 50 % (from 10 ‰ to 6 ‰). In the 950 to 1000 cm$^{-1}$ range, however, the effect is smaller. Although the same spectrometer and beam splitter was used in the right and left hand panel the channeling amplitudes as well as the reduction factor varies. This is due to wavelength dependent reflectivity of the beam splitter.

Moreover, this exercise demonstrates that a wedge of about 2° on the air gap eliminates channeling even without a larger wedge of the beam splitter substrate. However, such a spectrometer completely free of channeling would result in non-interchangeability with beam splitters having a smaller air gap wedge and therefore, the need to realign the spectrometer after a beam splitter exchange. Furthermore, when switching from small to large wedge two new matched beam splitters are needed since the KBr beam splitter does not transmit visible light and therefore a second one (normally CaF$_2$ or glass) is needed for the alignment procedure. Switching within this new pair of beam splitters is possible without realignment. The ILS of the spectrometers with such a pair of beam splitters is good.

## 6 Conclusions

Firstly, this paper documents the channeling amplitudes for nearly all of the FTIR spectrometers used in NDACC. Such a systematic performance analysis is needed for improving the trace gas retrievals and for calculating complete error budgets and also to improve the consistency and quality of the products across the NDACC network

Within NDACC, laboratory test spectra of about twenty spectrometers were recorded and analysed. The derived channeling amplitudes range from 0.1 to 2.0 ‰ and from 0.3 to 21 ‰ in the InSb and HgCdTe domains, respectively. These values are not negligible when constructing the error budget of minor trace gases. A reduction of the channeling amplitudes is highly desirable for the analysis of gases like ClONO$_2$, HNO$_3$, HCHO, and SF$_6$ since these species typically absorb in the order of about 5 ‰ (ClONO$_2$, HCHO) to 50 ‰ (HNO$_3$) of the incoming infrared light in the center of the signature.

Secondly, this study shows the potential to reduce channeling in several spectrometers and to improve the homogeneity within the network. The channeling frequencies allow us to determine the responsible optical component. A few instruments show channeling with a frequency of a few wavenumbers due to insufficiently wedged detector windows. Switching the detector window or, more easily, the entire detector including dewar and detector window, will help reduce channeling in these cases. Finally, we found that most spectrometers show two dominant channeling frequencies with about 0.1 or 0.2 cm$^{-1}$ and 0.9 cm$^{-1}$ corresponding to beam splitter substrate and beam splitter air gap, respectively, the latter usually dominant.. The option of reducing this channeling contribution was investigated by adjusting the wedge angles on a test beam splitter. Increasing the

wedge of this gap significantly reduces the channeling at 0.9 cm$^{-1}$ and therefore, such a beam splitter design offers the promise of further reducing channeling. As a result of this study, Bruker changed the standard air gap wedge of its beam splitters from 0.5° to 0.8°. Furthermore, beam splitters with a wedge of 2° are available on request. Switching to this modified beam splitter design would contribute to further homogenization of the spectrometers operated within NDACC.


**Appendix A**

**Table A1:** List of optical filters used in the IRWG (InfraRed Working Group) of NDACC.

| Filter number | Spectral range [μm] | Spectral range [cm$^{-1}$] | Target species examples |
|---|---|---|---|
| 1 | 2.2 - 2.6 | 3850 - 4550 | HF |
| 2 | 2.6 - 3.3 | 3030 - 3850 | HCN |
| 3 | 3.2 - 4.1 | 2440 - 3130 | HCl, $CH_4$, $C_2H_6$, HCHO, $NO_2$ |
| 4 | 3.9 - 5.0 | 2000 - 2560 | $N_2O$ |
| 5 | 4.6 - 6.3 | 1590 – 2170 | CO, NO, OCS |
| 6 | > 7.4 | < 1350 | $O_3$, $ClONO_2$, $HNO_3$, $SF_6$ |
| 7 | 9.8 – 13.0 | 770 - 1020 | $O_3$, $ClONO_2$, $HNO_3$ |
| 8 | 7.5 – 10.2 | 980 - 1330 | $O_3$ |

*Data availability.* Channeling test spectra used in this study are available on request from the corresponding author (thomas.blumenstock@kit.edu).

*Author contributions.* TB designed the study, performed the analysis, and wrote the paper. FH designed the analysis of the test spectra and wrote section 2 of the paper. AK improved the beam splitter and provided test spectra. All other authors did lab 370 measurements and provided test spectra. All authors read and provided feedback on the paper.

*Competing interests.* The authors declare no competing interests.

*Acknowledgements.* We acknowledge Gerhard Kopp for stimulating discussions on Fabry Perot fringing effects. The authors 375 like to acknowledge the project INMENSE (CGL2016-80688-P) funded by Ministerio de Economía y Competitividad from Spain. For the Altzomoni site UNAM (DGAPA grants IN111418 & IN107417), CONACYT (290589) and PASPA are acknowledged. The Paris site has received funding from Sorbonne Université, the French research center CNRS and the French space agency CNES. Operations at Rikubetsu and Tsukuba sites are supported in part by the GOSAT series project. SPbU

team was supported by Russian Foundation for Basic Research through the project no.18-05-00011. The Lauder and Arrival
Heights FTIR measurements are core-funded by NIWA through New Zealand's Ministry of Business, Innovation and
Employment Strategic Science Investment Fund. We also thank Antarctica New Zealand for providing support for the FTIR
measurements at Arrival Heights, which includes test spectra collection. The Jungfraujoch FTIR experiment has received
funding from the F.R.S. – FNRS, the Fédération Wallonie-Bruxelles, both in Brussels, Belgium, and from the GAW-CH
programme of MeteoSwiss. ULiège acknowledges that the International Foundation High Altitude Research Stations
Jungfraujoch and Gornergrat (HFSJG), 3012 Bern, Switzerland, made it possible to carry out our experiment at the
Jungfraujoch Station. We like to thank the AWI Bremerhaven and the personnel at the AWIPEV station, Ny-Ålesund
(Spitsbergen) for logistic and on-site support. Eureka measurements were made at the Polar Environment Atmospheric
Research Laboratory (PEARL), primarily supported by the Natural Sciences and Engineering Research Council of Canada
(NSERC), Environment and Climate Change Canada, and the Canadian Space Agency. Toronto measurements were made at
the University of Toronto Atmospheric Observatory (TAO), primarily supported by NSERC and the University of Toronto.
The National Center for Atmospheric Research is sponsored by the National Science Foundation. The NCAR FTS observation
programs at Thule, GR, Mauna Loa, HI and Boulder, CO are supported under contract by the National Aeronautics and Space
Administration (NASA). The FTIR stations Bremen, Garmisch, Izaña, Karlsruhe, and Ny-Ålesund have been supported by the
German Bundesministerium für Wirtschaft und Energie (BMWi) via DLR under grants 50EE1711A-B&D. This work has
been supported by the Federal Ministry of Education and Research (BMBF) Germany in the project TroStra (01LG1904A).

The article processing charges for this open-access publication were covered by a Research Centre of the Helmholtz
Association.

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
