# Peer review of "Characterisation and potential for reducing optical resonances in FTIR spectrometers of the Network for the Detection of Atmospheric Composition Change (NDACC)"

_Atmospheric Measurement Techniques, 2020_

## Referee Comment (RC1) · Anonymous Referee #1 · 6 Oct 2020

**General Comments.**
Good paper. Channel fringes are probably a major source of station-to-station bias within the NDACC-IRWG network, especially for weakly-absorbing gases. This is because the amplitude and phase of channel fringes can vary considerably from site to site, even for nominally-identical instruments. So the fringes must either be suppressed, or somehow accounted for in the spectral analysis, or both.

The main deficiency of this manuscript is that the authors provide no explanation of why increasing the wedge angle of the air-gap reduces the amplitude of channel fringes. The central conclusion of the paper is that an 0.8 deg angle for the air-wedge substantially reduces the channeling, as compared with the standard 0.5 deg. But the authors don't tell us why. Ideally, there would be an equation that relates the channel fringe amplitude to the relevant physical properties (reflectivity, flatness, wavenumber, wedge angle). This equation would also explain why the channel fringe amplitudes are so much larger in HgCd than in InSb. Alternatively, there should be a figure (fringe amplitude versus wedge angle for different wavenumbers) showing the results of computer modelling of the channeling.

In line 95 the authors further state (correctly) that a large tilt suppresses channel fringes, but don't offer any explanation why.

Wouldn't anti-reflection coating of the BS and Compensator also decrease the channeling? Explain why this isn't a feasible option?

Finally, the authors should discuss potential disadvantages of the larger wedge. For example, might an increased wedge angle between the BS and Compensator cause alignment problems for instruments that are aligned at 1 atm and then operated under vacuum? Or is the air-gap sealed such that the air pressure between the BS and Compensator never changes? Or is there another reason why this doesn't matter?

Paper should be publishable once these issues (above) are addressed. The authors should also address the more technical problems discussed below.

**Specific Comments.**
Line 40: "0.9 and 0.11 or 0.23" is ambiguous. I suggest two sentences, one describing air gap periods, and the second discussing the substrate periods.

Line 45: quantify "significantly"

Line 62: Here you use % as the unit of channel fringe amplitude. Is this a typo? In other places you use ‰. Choose a unit and be consistent.

Line 98: "design" → "build". It is easy to **design** an FTS free from channeling; just specify everything to be wedged.

Line 117: Explain why NDACC uses a set of filters (improve SNR and avoid saturation). This won't be apparent to a non-NDACC reader.

Line 117: here you use "arc-min" as the wedge angle unit, whereas previously you used degrees. Choose one and use consistently. Or explain why wedge angle requires two different units.

Line 121: I think that a table would be useful here (or a link to a table) that shows the spectral coverage of each NDACC filter. Also, add a column to Table 2 showing which filters were used at each site.

Line 130: "…spectral resolution of 0.05 cm-1" is ambiguous. Add the OPD parenthetically.

Line 148: Figure 2 caption inadequate.

Line 170: Move fig.3 earlier, before discussion of fig.4 begins.

Line 175: I don't think that the colors add much value to fig.3 since you've already told us the correspondence between the three optical cavities and their fringe periods. Perhaps add the HgCd information to fig. 3 and then use colors to denote the detector or the wavenumber of the fitted window.

Line 185: What about the fringes from the BS substrate? Are these never the largest?

Line 188: Site labels should be identical between figs.4 & 5 (IZ-18 vs IZ-2018)

Line 197: "The amplitude is even larger as compared to the InSb domain" → "The HgCd amplitudes are larger than those in the InSb domain". Explain why amplitude is larger in HdCd than in InSb domain?

Lines 200-203: Here you discuss the InSb domain in section 4.2 (HgCdTe Domain). Shouldn't these sentences be in section 4.1?

Line 204: Figure 6 doesn't explain what the three curves are. Are these spectra from different instruments? If so, which ones? Labelling the curves as weak/medium/strong isn't helpful. I can already see with my eyes which one has the strong fringes.

Line 208: Mixed units for wedge angles.

Line 211: "…with far-infrared pellicle…" →"…with unwedged far-infrared pellicle…"

Line 216: Fig.7 caption. What is the difference between upper and lower panels? Different instruments? If so, which ones? Are the left panels from the same instrument as the right panels? In the lower left panel increasing the wedge from 0.5 to 0.8 deg. caused a factor 3 reduction in the channel fringe amp. But in the lower-right panel, the reduction was much less, perhaps only a factor 1.5. Please discuss.

Line 218: "To avoid the need for strongly wedged substrates…". This is confusing since the surrounding discussion is about the air-gap fringes. A strongly wedged substrate won't change the air-gap wedge, unless there is an unspoken linkage between the two.

Line 232-233: As in line 218, here you mix the air-gap fringes and the substrate fringes. In my mind these are separate things, with different periods, controlled by different factors. So why would "a larger wedge of the beam slitter substrate" help reduce the air-gap channel fringes?

Line 234: Perhaps change "incompatibility" to "non-interchangeability"

Line 248: "Finally, we found that most spectrometers show two dominant channeling frequencies with about 0.1 or 0.2 cm$^{-1}$ and 0.9 cm$^{-1}$ corresponding to beam splitter substrate and beam splitter air gap. In most cases, the channeling caused by the gap of the beam splitter is the leading one. " → "Finally, we found that most spectrometers show two dominant channeling frequencies with about 0.1 or 0.2 cm$^{-1}$ and 0.9 cm$^{-1}$ corresponding to beam splitter substrate and beam splitter air gap, respectively, the latter usually dominant."

---

## Referee Comment (RC2) · Arndt Meier (Referee) · 18 Oct 2020

I wish to congratulate the authors on this well researched and well carried out scientific work.

The manuscript is well and clearly structured, concise, relevant, appropriately illustrated, easy to follow and demonstrates a very good command of the English language. A pleasure to read.

The work presented fits well within the scope of this journal. The work exposes, de-

scribes and resolves one of those nagging problems that have been a stone in the shoe of many researchers in this specialist field. The novelty and relevance lies primarily in discussing the issues caused by undesired optical resonances not from the perspective of an individual instrument but on a measurement network wide concise analysis and quantification of the variability and amplitude of these issues and how relevant these are to the overall error budget of trace gases reported by the NDACC (and TCCON) network. The authors include the principal manufacturer of the commonly used spectrometers (Bruker Optics) in the study. This is a good approach and a reflection of decades of good dialogue between cutting edge research and industry to mutual benefit. The authors also discuss and suggest practical technical solutions to the benefit of all affected.

The scientific work has been carried out diligently and the conclusions are sound and relevant. Proper credit is given to past investigations as well as the contributing community who are seemingly all included as co-authors. Abstract and title are appropriate and concise.

Below I have a short list of very minor comments and suggestions that the authors may wish to consider for the final version to improve clarity and readability, but it is nothing that should delay the publication of the final version even if left unconsidered.

- Page 3, section 2, Line 91 "Equation (1) is used to assign..." replace 'assign' with 'identify'

- Line 94 correct spelling is "a harmonic" (not an harmonic)

- Line 104, description of Figure 1: Consider adding "where 'l' is denoted 'd' in equation (1)"

- Line 137 " Then, the background was normalized and a straight line was subtracted using OriginTM software" How was the normalization carried out? Or did the authors mean to say ' Then, the background was normalized by subtracting a straight line (from

the laboratory spectra) using OriginTM software'?

- page 14, section 5, Line 235 Comment: I wouldn't stress this as an impediment. As long as no pellicle beam splitters are in use, and which seems to be the case for the NDACC (and I believe the TCCON as well) which are the focus of this study, there is no issue as long as the only or at best two beam splitters in use for a given instrument have the same air gap wedge of say 2 degrees. I'm not sure if an additional glass beam splitter is in use for the optical alignment of the FTS, in which case the same wedge would have to be used for that one, too.

- Line 238: "Such a systematic performance analysis is needed for improving the trace gas retrievals and for calculating complete error budgets." Comment: consider adding "also in order to improve the consistency and quality of the products across the NDACC network"

- Line 242 Comment: Perhaps a rough indication of typical relative absorption strengths of the weak absorbers listed by the authors would be helpful to put the channeling error amplitudes reported into perspective, possibly earlier in the discussion rather than here.

- Line 249 Consider replacing "leading one" with "dominating one" Given that Axel and Denis from Bruker Optics are among the co-authors it would be nice to have and indication (or ideally commitment) that beam splitters with a larger air gap of say 2 degrees are available as an option - if necessary at a small surcharge - for new orders or a modification service for existing beam splitters. That would be great to know even for users outside the NDACC community that may also be affected by channeling in their work.

Recalling a quote from the late Rodolphe "Rudy" Zander: "Bruker as a manufacturer may not be better or worse than other companies, but at least they are listening". I hope that this good spirit continues beyond the retirement of Axel who has been the link to the NDACC community for decades.

Thanks to all authors and contributors to this enlightening study.

---

## Referee Comment (RC3) · Anonymous Referee #3 · 3 Nov 2020

This paper describes the problem of optical interferences occurring in FTIR spectrometers that are used in NDACC: it describes some laboratory experiments aiming at identifying and characterising these interferences in about 25 of the NDACC FTIR spectrometers and attributes the interferences to the optical elements inside the spectrometers. It is shown that it is essentially the beamsplitter that causes these interferences. These interferences cause channeling in the spectra that make the observation of weak absorptions in atmospheric spectra difficult. The paper also shows test with beamsplitters with different wedges and concludes that beamsplitters (BS) with a wedge of the gap

between the BS and the compensator plate of 0.8° (instead of the actual standard 0.5°) would be a good choice to minimize the channeling and at the same time avoid re-alignments when exchanging BS.

General comments:

The paper is essentially a technical paper. It is very concise and reads easily; the objectives, methodology and conclusions are clearly formulated. However, being a technical paper, I have the feeling that some technical details are missing, or not clearly spelled out.

- Equation (1) provides the formula for the Free Spectral Range of a Fabry-Pérot (FP) etalon, but it is not mentioned how FSR is calculated for 'a resonator due to both substrates, the beamsplitter and the compensator plate' (line 165).

- Tables 3 and 4: at some sites, like Harestua, Garmisch, Altzomoni in Table 3, or Harestua, Zugspitze, Altzomoni in Table 4, some frequencies appear that are very different from the other ones, without any explanation as to their origin: are they due to window effects ? Why are some of these different frequencies classified in Table 2 as the 'standard' F2 or F3 frequencies ?

- In Table 4, A4 (= 21 pro mille) at Ny Alesund corresponds to F4 . Why is this amplitude included in the range of amplitudes of the channeling caused by the gap of the BS, with frequency F1 = 0.9 cm-1) ?

- In Table 4: why at Lauder, 2 different frequencies are assigned to F1 ? The same question holds for a few other sites and other frequencies (F2) in Table 4.

Specific comments:

- Line 49: I would specify 'total and partial column abundances' instead of simply 'column abundances'

- Line 93-94: The sentence is erroneous as it is formulated here. I suggest to replace

it as follows: "The Fabry-Pérot etalons generated by these optical components have rather low etendu and therefore the undesired parasitic effects caused in their spectral transmission is well described as an harmonic oscillation." I believe that this is what the authors intend to say. It would also be good to give the definition of 'etendu of a FP' here, or to add a reference to a definition.

- Table 1: Apparently the FSR given in the table assumes theta = 0°. However, in the standard NDACC FTIR spectrometer configuration, theta is typically 45° for the beamsplitter. So I am confused: how has the experiment been set up exactly ?

- Line 117: It is stated that NDACC filters with a wedge of 10', if properly oriented, do not cause channeling. Don't they cause any channeling at all, or are the frequencies of the channeling such that they don't disturb significantly the retrieval of typical NDACC atmospheric spectra ?

- Figure 2: Why has the x-axis been given in 1/Frequency whereas Figure 3 has an x-axis in frequency ?

The paper deserves being published, after some revisions to cope with the above comments.

---

## Author Comment (AC1) · 2 Dec 2020

**Response to comments from Referee 1**

Black: Referee's comments; Blue: Authors' answers

We thank referee #1 for the review and for providing useful feedback.

**Referee:**

**General Comments.**

Good paper.

Thank you very much!

Channel fringes are probably a major source of station-to-station bias within the NDACCIRWG network, especially for weakly-absorbing gases. This is because the amplitude and phase of channel fringes can vary considerably from site to site, even for nominally-identical instruments. So the fringes must either be suppressed, or somehow accounted for in the spectral analysis, or both.

The main deficiency of this manuscript is that the authors provide no explanation of why increasing the wedge angle of the air-gap reduces the amplitude of channel fringes. The central conclusion of the paper is that an 0.8 deg angle for the air-wedge substantially reduces the channeling, as compared with the standard 0.5 deg. But the authors don't tell us why. Ideally, there would be an equation that relates the channel fringe amplitude to the relevant physical properties (reflectivity, flatness, wavenumber, wedge angle). This equation would also explain why the channel fringe amplitudes are so much larger in HgCd than in InSb. Alternatively, there should be a figure (fringe amplitude versus wedge angle for different wavenumbers) showing the results of computer modelling of the channeling.

Section 2 on the background of the Fabry-Perot effect has been largely extended. Three examples are described in detail: a plane-parallel window at normal and 30° incidence and a wedged plate. Finally, the channeling amplitude as function of wedge angle was calculated and presented in an additional Figure (Fig. 2). These examples illustrate the wavelength dependence as well as the effect of wedging. While the channeling in a plane-parallel plate is not, the reduction by wedging the optical element is wavelength dependent.

In line 95 the authors further state (correctly) that a large tilt suppresses channel fringes, but don't offer any explanation why.

An explanation is added: Wedged optical components avoid channeling because the reflected beams do not superimpose and thus, do not interfere with each other.

Wouldn't anti-reflection coating of the BS and Compensator also decrease the channeling? Explain why this isn't a feasible option?

You're right, in principle an anti-reflection coating on the BS would decrease the channeling. However, such an AR coating is hardly compatible with the broad-band concept of beam splitters used in FTIR spectroscopy. Since the BS is specified for a very large spectral range, for example from 700 to 5000 cm$^{-1}$ for the KBr, such an AR coating would be very complex and would consist of several layers. It is very hard to completely suppress reflections for the entire spectral range without adding any undesirable effects like absorptions or reflections within this muliti-layer coating.

Finally, the authors should discuss potential disadvantages of the larger wedge. For example, might an increased wedge angle between the BS and Compensator cause alignment problems for instruments that are aligned at 1 atm and then operated under vacuum? Or is the air-gap sealed such that the air pressure between the BS and Compensator never changes? Or is there another reason why this doesn't matter?

The air gap is not sealed. In fact, there is a tiny difference in alignment under vacuum. However, this little difference occurs in instruments with small as well as with large BS wedge. So, at least part of this difference has another reason.

The disadvantage of the larger wedge is its incompatibility with other beam splitters. Of course, it is not compatible with pellicle BS used in the FIR spectral domain. Besides this, switching from small to large wedge is quite an effort since two new beam splitters are needed. The KBr BS does not transmit visible light and therefore a second BS (normally $CaF_2$ or glass) is needed for the alignment procedure by which interference fringes are checked by eye or camera. Furthermore, a full alignment of the spectrometer is needed when switching from small to large wedge. The alignment procedure recommended in NDACC is an effort and described in http://www.acom.ucar.edu/irwg/Griffith_alignment.pptx and https://www.acom.ucar.edu/irwg/HaseBlumenstockAlignment.pdf.

For new instruments switching to a larger wedge is easier since the spectrometer is aligned by the manufacturer. However, a BS with large wedge is not listed in the price list and is available on request only. And the company asks for orders of several items at the same time. At least for new customers or customers from outside the NDACC community this might be hard to know and to order correctly.

Once switched to a pair of beam splitters with increased and matched wedge there is no disadvantage. Of course, the spectrometer needs re-alignment when switching to a larger wedge. This has been done at several sites (Altzomoni, Izaña, Karlsruhe and Kiruna) and is working fine. Switching within this new pair of beam splitters is possible without realignment. The ILS of these instruments is good.

The disadvantages of the larger wedge are discussed in lines 233ff. A few sentences were added here.

Paper should be publishable once these issues (above) are addressed. The authors should also address the more technical problems discussed below.

**Specific Comments.**

Line 40: "0.9 and 0.11 or 0.23" is ambiguous. I suggest two sentences, one describing air gap periods, and the second discussing the substrate periods.

Done.

Line 45: quantify "significantly"

A sentence is added to quantify the reduction of channeling amplitude with increasing wedge of the air gap.

Line 62: Here you use % as the unit of channel fringe amplitude. Is this a typo? In other places you use ‰. Choose a unit and be consistent.

Done. It was not a typo. We thought % is more appropriate in the introduction to give the magnitude of the effect while ‰ is more appropriate to give the exact numbers in the result section. Anyway, ‰ is used consistently throughout the paper.

Line 98: "design" à "build". It is easy to design an FTS free from channeling; just specify everything to be wedged.

Changed. Well, some devices are difficult to wedge, for example pellicle beam splitters or detector elements. The latter might also cause channeling. Finally, it depends on the wavelength as pointed out in chapter 2. In the NIR spectral domain you're right. In the FIR or even millimeter wave region, however, channel free instruments might be even hard to design.

Line 117: Explain why NDACC uses a set of filters (improve SNR and avoid saturation). This won't be apparent to a non-NDACC reader.

Done.

Line 117: here you use "arc-min" as the wedge angle unit, whereas previously you used degrees. Choose one and use consistently. Or explain why wedge angle requires two different units.

Units were taken from the data sheet of the manufacturer. Changed to degrees.

Line 121: I think that a table would be useful here (or a link to a table) that shows the spectral coverage of each NDACC filter. Also, add a column to Table 2 showing which filters were used at each site.

Done: A column is added to Table 2 and a table of the NDACC filters is added in Appendix A.

Line 130: "…spectral resolution of 0.05 cm$^{-1}$" is ambiguous. Add the OPD parenthetically.

Done.

Line 148: Figure 2 caption inadequate.

Modified.

Line 170: Move fig.3 earlier, before discussion of fig.4 begins.

Done.

Line 175: I don't think that the colors add much value to fig.3 since you've already told us the correspondence between the three optical cavities and their fringe periods. Perhaps add the HgCd information to fig. 3 and then use colors to denote the detector or the wavenumber of the fitted window.

Done.

Line 185: What about the fringes from the BS substrate? Are these never the largest?

You're right, there is one case: At Rikubetsu, the substrate of the KBr beam splitter causes the largest channeling amplitude. Fig. 5 has been changed accordingly.  For the CaF$_2$ beam splitter there is no such case (line 185, Fig. 4).

Line 188: Site labels should be identical between figs.4 & 5 (IZ-18 vs IZ-2018)

Done.

Line 197: "The amplitude is even larger as compared to the InSb domain" à "The HgCd amplitudes are larger than those in the InSb domain". Explain why amplitude is larger in HdCd than in InSb domain?

Done. Is explained in Sect. 2. See also comment and additions to chapter 2.

Lines 200-203: Here you discuss the InSb domain in section 4.2 (HgCdTe Domain). Shouldn't these

sentences be in section 4.1?

Fig. 6 as well as lines 200-203 present and discuss results of the HgCdTe domain. In line 200

(all line numbers refer to original version) 'HgCdTe spectra' was added for clarity.

Line 204: Figure 6 doesn't explain what the three curves are. Are these spectra from different

instruments? If so, which ones? Labelling the curves as weak/medium/strong isn't helpful. I can already see with my eyes which one has the strong fringes.

The spectra are taken from different instruments. Labelling of the curves is changed.

The idea of this figure was just to visualize the range of channeling amplitudes within the NDACC network.

Line 208: Mixed units for wedge angles.

Done.

Line 211: "…with far-infrared pellicle…" à"…with unwedged far-infrared pellicle…"

Done.

Line 216: Fig.7 caption. What is the difference between upper and lower panels? Different instruments?

If so, which ones? Are the left panels from the same instrument as the right panels? In the lower left panel increasing the wedge from 0.5 to 0.8 deg. caused a factor 3 reduction in the channel fringe amp. But in the lower-right panel, the reduction was much less, perhaps only a factor 1.5. Please discuss.

These measurements were all made with the same instrument. All measurements of Fig. 7 (Fig. 8 in the revised version) were made at Bruker company in Ettlingen. In the first setup, beam splitter with 0.5°, 1.2° and 2.2° were tested (upper panel). The beam splitter was the same for all 3 angles, just different spacers were used.

Since a wedge of 0.8° was chosen for standard beam splitters a test with 0.5° and 0.8° wedge was conducted later on (lower panel). This setup used the same spectrometer as compared to the previous setup (upper panel). The beam splitter is the same for the right and left panel.

Spectra shown in the right and left hand panel show different spectral regions. The channeling amplitudes as well as the reduction factor varies presumably due to wavelength dependent reflectivity of the beam splitter.

Line 218: "To avoid the need for strongly wedged substrates…". This is confusing since the surrounding discussion is about the air-gap fringes. A strongly wedged substrate won't change the air-gap wedge, unless there is an unspoken linkage between the two.

Yes, the strongly wedged substrate won't change the air gap. The idea of line 218 is to make clear that the following paragraph is on air gap fringes only. The discussion before line 218 (line 206-212) is on a special BS with larger wedge of the substrate and of the air gap.

Line 232-233: As in line 218, here you mix the air-gap fringes and the substrate fringes. In my mind these are separate things, with different periods, controlled by different factors. So why would "a larger wedge of the beam slitter substrate" help reduce the air-gap channel fringes?

We agree that a larger wedge of the substrate does not reduce the air gap channel fringes. However, some spectrometer do also show channeling of the substrate. And therefore, in a much earlier attempt to reduce channeling, the air gap and the substrate were strongly wedged at some sites (line 206-212). This chapter is on reducing the channeling of the entire beam splitter not only of the air gap channel. Line 232f is a kind of a summary of chapter 5 highlighting the result of this study that it is possible to manufacture a beam splitter free of 0.11, 0.23 and 0.9 cm$^{-1}$ channel fringes even with a small wedge of the substrate.

Line 234: Perhaps change "incompatibility" to "non-interchangeability"

Done.

Line 248: "Finally, we found that most spectrometers show two dominant channeling frequencies with about 0.1 or 0.2 cm$^{-1}$ and 0.9 cm$^{-1}$ corresponding to beam splitter substrate and beam splitter air gap. In most cases, the channeling caused by the gap of the beam splitter is the leading one. " à "Finally, we found that most spectrometers show two dominant channeling frequencies with about 0.1 or 0.2 cm$^{-1}$ and 0.9 cm$^{-1}$ corresponding to beam splitter substrate and beam splitter air gap, respectively, the latter usually dominant."

Done. Thank you for the corrections!

---

## Author Comment (AC2) · 2 Dec 2020

**Response to comments from Referee 2 (Dr. Arndt Meier)**

Black: Referee's comments; Blue: Authors' answers

We thank referee #2, Dr. Arndt Meier, for the review and for support of the paper.

**Referee:**

I wish to congratulate the authors on this well researched and well carried out scientific work.
The manuscript is well and clearly structured, concise, relevant, appropriately illustrated, easy to follow and demonstrates a very good command of the English language.
A pleasure to read.

Thank you very much!

The work presented fits well within the scope of this journal. The work exposes, describes and resolves one of those nagging problems that have been a stone in the shoe of many researchers in this specialist field. The novelty and relevance lies primarily in discussing the issues caused by undesired optical resonances not from the perspective of an individual instrument but on a measurement network wide concise analysis and quantification of the variability and amplitude of these issues and how relevant these are to the overall error budget of trace gases reported by the NDACC (and TCCON) network. The authors include the principal manufacturer of the commonly used spectrometers (Bruker Optics) in the study. This is a good approach and a reflection of decades of good dialogue between cutting edge research and industry to mutual benefit. The authors also discuss and suggest practical technical solutions to the benefit of all affected.

The scientific work has been carried out diligently and the conclusions are sound and relevant. Proper credit is given to past investigations as well as the contributing community who are seemingly all included as co-authors. Abstract and title are appropriate and concise.

Below I have a short list of very minor comments and suggestions that the authors may wish to consider for the final version to improve clarity and readability, but it is nothing that should delay the publication of the final version even if left unconsidered.

- Page 3, section 2, Line 91 "Equation (1) is used to assign..." replace 'assign' with 'identify'

Done.

- Line 94 correct spelling is "a harmonic" (not an harmonic)

Corrected.

- Line 104, description of Figure 1: Consider adding "where 'l' is denoted 'd' in equation (1)"

Done.

- Line 137 " Then, the background was normalized and a straight line was subtracted using OriginTM software" How was the normalization carried out? Or did the authors mean to say ' Then, the background was normalized by subtracting a straight line (from the laboratory spectra) using OriginTM software'?

Is clarified in the text and in the caption of Fig. 2:
The background was normalized by dividing a straight line that connects the ends of the spectrum using ORIGIN TM software (red line in Fig.2a). The resulting quotient minus 1 (Fig. 2b) was used to perform a FFT analysis.

- page 14, section 5, Line 235 Comment: I wouldn't stress this as an impediment. As long as no pellicle beam splitters are in use, and which seems to be the case for the NDACC (and I believe the TCCON as well) which are the focus of this study, there is no issue as long as the only or at best two beam splitters in use for a given instrument have the same air gap wedge of say 2 degrees. I'm not sure if an additional glass beam splitter is in use for the optical alignment of the FTS, in which case the same wedge would have to be used for that one, too.

Correct, for the standard alignment procedure a second beam splitter (CaF$_2$ or glass) is needed to observe Haidinger fringes with a telescope. We agree that exclusion of a pellicle beam splitter is not a show stopper for the NDACC and TCCON community, at least when purchasing a new instrument.
New colleagues buying a new instrument might not know this option. For existing instruments, however, switching to beam splitters with larger wedge means an investment of two beam splitters and moreover, a full re-alignment of the spectrometer!

- Line 238: "Such a systematic performance analysis is needed for improving the trace gas retrievals and for calculating complete error budgets." Comment: consider adding "also in order to improve the consistency and quality of the products across the NDACC network"

Done.

- Line 242 Comment: Perhaps a rough indication of typical relative absorption strengths of the weak absorbers listed by the authors would be helpful to put the channeling error amplitudes reported into perspective, possibly earlier in the discussion rather than here.

Added.

- Line 249 Consider replacing "leading one" with "dominating one"

Replaced.

Given that Axel and Denis from Bruker Optics are among the co-authors it would be nice to have And indication (or ideally commitment) that beam splitters with a larger air gap of say 2 degrees are available as an option - if necessary at a small surcharge - for new orders or a modification service for existing beam splitters. That would be great to know even for users outside the NDACC community that may also be affected by channeling in their work.

Agreed. Since recently and as a result of this study the standard air wedge of Bruker beam splitters is 0.8° instead of 0.5°. Beam splitters with an air wedge of 2 degrees are available on request if there is

a joint order of a sufficient number of pieces. Up to now this item (beam splitter with 2° air gap) is not included in the price list and is available on request only. A modification service is also available. We agree this option is hard to know for users outside the NDACC community or for newcomers. Therefore, a sentence on availability has been added.

---

## Author Comment (AC3) · 2 Dec 2020

**Response to comments from Referee 3**

Black: Referee's comments; Blue: Authors' answers

We thank referee #3 for the review and for providing useful feedback.

**Referee:**

This paper describes the problem of optical interferences occurring in FTIR spectrometers that are used in NDACC: it describes some laboratory experiments aiming at identifying and characterising these interferences in about 25 of the NDACC FTIR spectrometers and attributes the interferences to the optical elements inside the spectrometers. It is shown that it is essentially the beamsplitter that causes these interferences. These interferences cause channeling in the spectra that make the observation of weak absorptions in atmospheric spectra difficult. The paper also shows test with beamsplitters with different wedges and concludes that beamsplitters (BS) with a wedge of the gap between the BS and the compensator plate of 0.8_ (instead of the actual standard 0.5_) would be a good choice to minimize the channeling and at the same time avoid re-alignments when exchanging BS.

General comments:
The paper is essentially a technical paper. It is very concise and reads easily; the objectives, methodology and conclusions are clearly formulated. However, being a technical paper, I have the feeling that some technical details are missing, or not clearly spelled out.
- Equation (1) provides the formula for the Free Spectral Range of a Fabry-Pérot (FP) etalon, but it is not mentioned how FSR is calculated for 'a resonator due to both substrates, the beamsplitter and the compensator plate' (line 165).

The formula calculates the resulting frequency out of the cavity length. Here, it is used the other way around. The observed channeling frequency is used to calculate the optical thickness. A channeling frequency of 0.11 $cm^{-1}$ corresponds to 30 mm of KBr which includes beam splitter and compensator plate.

- Tables 3 and 4: at some sites, like Harestua, Garmisch, Altzomoni in Table 3, or Harestua, Zugspitze, Altzomoni in Table 4, some frequencies appear that are very different from the other ones, without any explanation as to their origin: are they due to window effects ? Why are some of these different frequencies classified in Table 2 as the 'standard' F2 or F3 frequencies ?

Yes, this kind of channeling is caused by the detector window.
Line 166f states 'A few spectrometers show an additional channeling fringe with a frequency of about 3 $cm^{-1}$. This is due to the detector window that is often made of sapphire or calcium fluoride ($CaF_2$).' And similar in line 193f:
'Two spectrometers show an additional channeling frequency of 2.17 and 3.85 $cm^{-1}$, indicating that the wedge of the detector window is not sufficient in these cases.'
We added the site names to clarify this.
Please also note the color code in Figs. 4 and 5 to indicate the origin of the channeling with the largest amplitude at each site.

Table 2 or 3 (We guess you refer to Table 3):

If the standard F2 or F3 frequency was not observed other frequencies moved forward (Toronto, Harestua, Garmisch, Zugspitze and Altzomoni).

- In Table 4, A4 (= 21 pro mille) at Ny Alesund corresponds to F4 . Why is this amplitude included in the range of amplitudes of the channeling caused by the gap of the BS, with frequency F1 = 0.9 cm$^{-1}$) ?

Yes, F4 (2.17 cm$^{-1}$) is attributed to an optical window, for example the detector window, see line 193-194. Also in Fig. 5 color code denotes F4 as caused by a window (in blue) not the gap of the BS.

The range covers the entire range observed not only due to BS channeling. For clarity, the sequence of sentences has been changed in line 195f: Instead of 'Figure 5 shows the amplitude of the strongest channeling frequency of each spectrometer. In most cases, channeling caused by the gap of the beam splitter is the most pronounced one. The amplitudes range from 0.3 to 21 ‰ with …' has been changed to 'Figure 5 shows the amplitude of the strongest channeling frequency of each spectrometer. The amplitudes range from 0.3 to 21 ‰ with … . In most cases, channeling caused by the gap of the beam splitter is the most pronounced one.'
And similar in line 170-172.

- In Table 4: why at Lauder, 2 different frequencies are assigned to F1 ? The same question holds for a few other sites and other frequencies (F2) in Table 4.

If the FFT analysis yields channeling at two frequencies close to each other the corresponding amplitudes were listed in the same column. You're right, for Lauder the second frequency in the F1 column does not fit here. It is changed in Table 4.

Specific comments:
- Line 49: I would specify 'total and partial column abundances' instead of simply 'column abundances'

Done.

- Line 93-94: The sentence is erroneous as it is formulated here. I suggest to replace it as follows: "The Fabry-Pérot etalons generated by these optical components have rather low etendu and therefore the undesired parasitic effects caused in their spectral transmission is well described as an harmonic oscillation." I believe that this is what the authors intend to say. It would also be good to give the definition of 'etendu of a FP' here, or to add a reference to a definition.

Sentence is corrected as suggested. Thanks for pointing to this paragraph and we also found a mix-up of terminology: we intended to refer to the finesse of the resonator here, not to the etendue.  The low reflectivity yields a low finesse. The finesse is a measure of the number of reflections within a cavity and a low reflectivity means a low finesse and small number of reflections within the cavity.

- Table 1: Apparently the FSR given in the table assumes theta = 0_. However, in the standard NDACC FTIR spectrometer configuration, theta is typically 45_ for the beamsplitter. So I am confused: how has the experiment been set up exactly ?

You're right, the FSR given in the table is calculated with theta = 0.  In the NDACC FTIR spectrometer configuration theta is 30°. However, due to refraction theta is smaller inside the beam splitter. According to Snell's law theta is 19° for $n$=1.5. Cos 19° is about 0.95 and therefore close to 1.0.

- Line 117: It is stated that NDACC filters with a wedge of 10', if properly oriented, do not cause channeling. Don't they cause any channeling at all, or are the frequencies of the channeling such that they don't disturb significantly the retrieval of typical NDACC atmospheric spectra ?

If the wedge is sufficient they don't cause any channeling at all. The reflecting beams do not overlap and thus do not interfere with each other.

- Figure 2: Why has the x-axis been given in 1/Frequency whereas Figure 3 has an x-axis in frequency ?

In the OriginTM software the inverse FFT has been applied which calculates the results as function of 1/frequency as shown in Fig. 2. For the presentation and discussion of the results the results were given in terms of frequency to be consistent with the spectra shown in Figs. 6 and 7.

The paper deserves being published, after some revisions to cope with the above comments.